# Autoimmune Hemolytic Anemia in the Pediatric Setting

**DOI:** 10.3390/jcm10020216

**Published:** 2021-01-09

**Authors:** Aikaterini Voulgaridou, Theodosia A. Kalfa

**Affiliations:** 1Division of Hematology, Cancer and Blood Diseases Institute, Cincinnati Children’s Hospital Medical Center, Cincinnati, OH 45229, USA; Aikaterini.Voulgaridou@cchmc.org; 2Department of Pediatrics, University of Cincinnati College of Medicine, Cincinnati, OH 45267, USA

**Keywords:** warm autoimmune hemolytic anemia, cold agglutinin syndrome, paroxysmal cold hemoglobinuria, direct antiglobulin test

## Abstract

Autoimmune hemolytic anemia (AIHA) is a rare disease in children, presenting with variable severity. Most commonly, warm-reactive IgG antibodies bind erythrocytes at 37 °C and induce opsonization and phagocytosis mainly by the splenic macrophages, causing warm AIHA (w-AIHA). Post-infectious cold-reactive antibodies can also lead to hemolysis following the patient’s exposure to cold temperatures, causing cold agglutinin syndrome (CAS) due to IgM autoantibodies, or paroxysmal cold hemoglobinuria (PCH) due to atypical IgG autoantibodies which bind their target RBC antigen and fix complement at 4 °C. Cold-reactive antibodies mainly induce intravascular hemolysis after complement activation. Direct antiglobulin test (DAT) is the gold standard for AIHA diagnosis; however, DAT negative results are seen in up to 11% of warm AIHA, highlighting the need to pursue further evaluation in cases with a phenotype compatible with immune-mediated hemolytic anemia despite negative DAT. Prompt supportive care, initiation of treatment with steroids for w-AIHA, and transfusion if necessary for symptomatic or fast-evolving anemia is crucial for a positive outcome. w-AIHA in children is often secondary to underlying immune dysregulation syndromes and thus, screening for such disorders is recommended at presentation, before initiating treatment with immunosuppressants, to determine prognosis and optimize long-term management potentially with novel targeted medications.

## 1. Introduction

Autoimmune hemolytic anemia (AIHA) is an acquired form of hemolytic anemia in which autoantibodies target red blood cell (RBC) membrane antigens, inducing cell rupture (lysis). Hemolysis triggers compensatory RBC production by increasing erythropoietin levels; however, this response is typically insufficient to redress normal hemoglobin blood levels leading to anemia. AIHA is characterized as “extrinsic” because the autoantibodies affect otherwise normal RBCs. A recent systematic review assessing AIHA terminology concluded that there is significant heterogeneity in the definition and diagnostic criteria for the disease [1]. In most of the reviewed studies, AIHA was defined as hemolytic anemia with a positive direct antiglobulin test (DAT) and concurrent exclusion of alternative diagnoses. However, there are limitations in the use of that definition since it does not include DAT-negative cases. AIHA is classified as “warm” or “cold” based on the optimal temperature at which the antibodies present maximal reactivity and as primary or secondary depending on the presence of a recognized underlying cause, such as immunodeficiency, infections, medications, or malignancy [2]. It affects both pediatric and adult populations, although its presentation in childhood is relatively rare, with the annual incidence estimated to be approximately 0.8 per 100,000 individuals under 18 years old [3].

Children with AIHA can present with a variable degree of severity. The most common form of AIHA in the pediatric population is due to warm-reactive autoantibodies. Warm antibody AIHA (w-AIHA) is diagnosed in more than half of autoimmune hemolytic episodes [4,5,6]. Cold reactive antibodies are responsible for the less frequent forms of the disease, known as cold agglutinin syndrome (CAS) and paroxysmal cold hemoglobinuria (PCH), defined by the immunoglobulin (Ig) isotype against the RBCs: IgM in CAS and IgG in PCH. A small subset of cases is recognized as “mixed AIHA,” with laboratory work-up revealing serologic findings of both w-AIHA and CAS (Table 1).

An acute presentation of autoimmune hemolytic anemia is frequently a life-threatening, fast-progressive disease and requires prompt diagnosis, initiation of treatment, and close monitoring. Therefore, the first question to be answered on presentation of a patient with evidence of hemolytic anemia, is if this is an immune-mediated hemolytic anemia by ruling out other potential causes (Table 2). To resolve this question, based on an algorithmic approach for the evaluation of hemolytic anemia, the DAT and indirect antiglobulin test (IAT), historically known as direct and indirect Coombs respectively, are the first tests to be ordered, along with complete blood count (CBC), reticulocyte count, and blood smear preparation for review. An IAT in most blood bank and laboratory systems is orderable as the “antibody screen,” obtained when a “Type + Screen” order is placed.

We want to emphasize here that although DAT is the gold standard for diagnosis of AIHA, test limitations exist. In DAT, a polyspecific anti-IgG and anti-C3 reagent, is added to the patient’s washed RBCs in suspension, resulting in cell agglutination when positive [7]. Then, the monospecific reagents anti-IgG and anti-C3d are used separately to detect IgG and complement, respectively. It is estimated that DAT is negative in up to 11% of all w-AIHA cases with clinical characteristics of autoimmune hemolysis, termed “DAT-negative w-AIHA” [5,8,9]. Awareness of the limitations of the (conventional) DAT assay performed in most laboratories and blood banks is needed, to pursue further evaluation in cases that appear compatible with immune-mediated hemolytic anemia despite DAT being negative. Enhanced DAT assays, often called super-Coombs, are available in reference laboratories and significantly increase the true-positive detection rate [10].

## 2. Warm Antibody AIHA (w-AIHA)

As an atypical but instructive case, a 12-year-old boy was referred to our clinic for a second opinion. He had been followed for the previous year at his local pediatric hematology-oncology center with hemolytic anemia. At the time of his initial presentation, he had anemia, fatigue, jaundice, and splenomegaly. Laboratory values were indicative of hemolytic anemia, with hemoglobin (Hgb) down to 73 g/L, elevated reticulocyte count at 5.5%, increased lactate dehydrogenase (LDH) at 561 units/L (normal range 100–325 units/L) and uric acid at 10.1 mg/dL (normal range 3.4–6.9 mg/dL). DAT was negative. He was transfused with a packed RBC (PRBC) unit which he tolerated well, but he had a suboptimal response of hemoglobin increase to 80 g/L. Since the significant splenomegaly and increased uric acid was concerning for malignancy such as lymphoma, he underwent computed tomography (CT) of his chest, abdomen, and pelvis, revealing hepatosplenomegaly and several mesenteric lymph nodes. He had a negative positron emission tomography (PET) scan and he also had some of the abdominal lymph nodes removed for pathology review that revealed benign reactive hyperplasia.

Evaluation for hereditary hemolytic anemia (HHA) with a next-generation sequencing (NGS) panel for RBC membrane disorders, RBC enzymopathies, and congenital dyserythropoietic anemias was pursued which did not reveal any pathogenic gene variants. In our clinic, about a year after initial presentation, he had compensated hemolysis with Hgb of 120 g/L and 5.7% reticulocytes, absolute reticulocyte count (ARC) of 214 × 10^9^/L, although with massive splenomegaly. Flow cytometry for paroxysmal nocturnal hemoglobinuria (PNH) was negative, and an ultrasound of the abdomen with doppler confirmed a significantly enlarged spleen with a longitudinal diameter of 19.3 cm, with no evidence of thrombosis of the portal or splenic vein and normal flow in all hepatic vessels. Blood smear review was remarkable for polychromasia and occasional spherocytes and microspherocytes (Figure 1). Osmotic gradient ektacytometry was compatible with spherocytosis showing mildly increased Omin and decreased EImax [11], a picture that can be seen with w-AIHA when a significant percentage of the patient’s RBCs lose membrane surface area and become spherocytes. The super-Coombs test was sent to a referral testing center which detected low-affinity IgG on the patient’s RBCs when the cells were washed at 4 °C with low ionic strength solution (LISS), demonstrating his diagnosis as “(conventional) DAT-negative” w-AIHA. An autoimmune lymphoproliferative syndromes (ALPS) screening panel was positive with a score of 4/4, and follow-up sequencing on the ALPS gene panel (including the genes *CASP8, CASP10, FADD, FAS, FASLG, ITK, KRAS, MAGT1, NRAS)* was conducted and detected a variant of uncertain significance (VUS) in *FAS*: c.710C > T (p.A237V). Since his hemolysis was fairly well compensated, the patient was started on sirolimus without concurrent steroid treatment for his w-AIHA, with resolution of hemolysis by the time of follow-up 4 months later and no palpable splenomegaly after a year.

This case illustrates several points worth noting regarding w-AIHA in childhood: while it is prudent to consider other possible causes of hemolysis, the clinical suspicion of AIHA with a new-onset hemolytic anemia has to be pursued further with an enhanced DAT assay. In a few cases of DAT-negative w-AIHA, even the enhanced DAT may be negative. In such cases, a therapeutic trial with intravenous immunoglobulin (ivIg) may need to be considered to demonstrate the immune-mediated etiology of the patient’s hemolytic anemia. Moreover, atypical w-AIHA and/or other autoimmune cytopenias have to trigger consideration for underlying immunodeficiencies or immune dysregulation syndromes, such as ALPS or ALPS-like disorders.

A negative DAT result can be due to several factors, such as low-affinity RBC autoantibodies that are easily removed from the RBC membrane during cell washings. In the case described above, the DAT negativity was overcome by doing the cell washes in LISS at 4 °C, allowing detection of the causative antibodies. Conventional DAT assays also fail to identify immunoglobulin subtypes other than IgG; occasionally, w-AIHA may be caused by IgA or warm-reactive monovalent IgM autoantibodies. In those cases, the autoantibodies may be detected by enhanced DAT assays utilizing reagents such as anti-IgA and anti-IgM to achieve agglutination of such antibody-coated erythrocytes or by flow cytometry. IgG autoantibodies may bind to the RBCs at a relatively low level, causing hemolysis, but being below the threshold of detection for the commonly performed DAT, while flow cytometry is more sensitive to detect these low levels of antibodies bound to RBCs [12].

### 2.1. Pathophysiology and Underlying Etiology

The warm-reactive antibodies causing AIHA bind to the RBC membrane antigens at 37 °C and are typically IgG; IgA and monomeric IgM are detected in rare cases [13]. The autoantibodies are polyclonal and poly-specific, i.e., they react with multiple RBC antigens rather than a specific one and are usually directed against high incidence antigens. Hemolysis is mainly extravascular, as RBCs coated with warm-reactive antibodies are phagocytosed by splenic macrophages carrying Fcg receptors. When the antibodies on the RBC membrane have high concentration or high affinity for complement, they trigger the activation cascade up to C3b and the C3b-opsonised erythrocytes are phagocytosed by the liver macrophages carrying C3-receptors. Rarely, the complement activation may proceed to the formation of membrane attack complex (C5b9), resulting in intravascular hemolysis [14].

A significant percentage of w-AIHA in children is primary or idiopathic; in a nationwide French cohort study of 265 children, that percentage approached 40% [4]. Underlying disorders leading to secondary w-AIHA most commonly include immunodeficiencies such as common variable immunodeficiency (CVID) and ALPS or other ALPS-like syndromes, autoimmune diseases such as systemic lupus erythematosus (SLE) and juvenile idiopathic arthritis, and infections, mostly viral [4]. Less frequent causes include malignancies, previous transfusions or transplantation, especially when treated with tacrolimus for post-transplant immunosuppression [15], and medications such as cephalosporins and piperacillin [16].

Evans syndrome accounts for up to 30% of AIHA in childhood and is characterized by the concurrent presence of at least two immune cytopenias. AIHA may coexist with immune thrombocytopenia (ITP) and/or autoimmune neutropenia (AIN) or it may precede or follow in clinical presentation ITP or AIN for months or years [4,17]. Of note, ITP and AIN are now recognized as Evans syndrome, even without association with AIHA. A range of immunoregulatory disorders have been related to Evans syndrome, triggering its pathology [18]. In a large cohort study, 80 patients with Evans syndrome underwent genetic testing searching for mutations in the most common genes involved in the pathogenesis of immunodeficiencies; 52 of them (65%) received a genetic diagnosis determining a pathogenic (*n* = 32, 40%) or probably pathogenic (*n* = 20, 25%) variant [19]. In ALPS, apoptosis-resistant and autoreactive lymphocytes can cause severe autoimmune dysregulation and lead to refractory Evans syndrome. With the increasing recognition of underlying immunodeficiencies for w-AIHA, it is highly recommended that all children presenting with Evans syndrome are screened for ALPS, CVID, as well as for HIV, especially in adolescents. Close follow-up and regular re-examinations are also important, as multi-lineage autoimmune cytopenias can be the single initial sign of SLE, with other SLE manifestations presenting later in the course of the disease [20].

Pediatric w-AIHA associated with giant cell hepatitis (GCH) deserves special mention. This association is a rare and distinct entity that usually presents in infancy and as early as in the neonatal period [21]. Initial clinical and laboratory findings are derived from AIHA and include jaundice, hepatosplenomegaly, and a positive Coombs test. Signs of liver disease, such as cholestasis, jaundice, and elevated aminotransferases, can follow those of hemolysis within a few days to several months [22]. Liver ultrasound is typically negative for parenchymal lesions, while serologic testing for viral, metabolic, and common autoimmune causes of hepatitis, such as antinuclear, anti-mitochondrial, and anti-smooth muscle antibodies, is negative [23]. The diagnosis is confirmed by liver biopsy, demonstrating prominent distortion of the hepatic architecture, diffuse giant cell transformation, and central-portal bridging fibrosis [23].

GCH with AIHA has a poor prognosis ending in fulminant hepatitis without aggressive treatment [21]. Treating those patients is challenging and early management is crucial to prevent irreversible liver failure [24]. Relapses often occur and complete remission is difficult to obtain [25,26]. Additionally, those cases have an increased risk of severe to fatal infections [24]. Although the pathogenesis of the disease is unclear, some authors support the implication of exaggerated B-cell activity due to the favorable response of those cases to anti-CD20 antibodies [27,28]. Additional prolonged immunosuppression with azathioprine is also used [27,29]. Thus, more studies on both the underlying mechanism of the disease and successful treatment strategies are required. Nevertheless, based on the severity of GCH with AIHA, liver function tests should be conducted in every infant diagnosed with AIHA regularly, regardless of its low incidence.

### 2.2. Clinical and Laboratory Findings

Clinical presentation of w-AIHA involves non-specific signs and symptoms of jaundice, dark urine, fatigue, splenomegaly and possibly hepatomegaly, and, in chronically persistent cases, cholelithiasis and cholecystitis [30]. In children with secondary w-AIHA, clinical manifestations of the underlying disorder may be present, as well.

The initial laboratory tests for w-AIHA aim to evaluate the presence of hemolysis and degree of anemia and include CBC with differential, reticulocyte count, peripheral blood smear review, DAT, type and screen, and serum markers such as total and unconjugated bilirubin, LDH, and haptoglobin. In most patients with hemolysis, low hemoglobin levels, elevated reticulocyte count, and depleted haptoglobin are detected. Elevated LDH and unconjugated bilirubin are common findings, as well. A review of the peripheral smear often illustrates the presence of microspherocytes and polychromasia, as in Figure 1a [31]. Not infrequently, reticulocytopenia may be noted with w-AIHA, either early in the course of the disease before reticulocyte response is triggered, or when the anti-RBC antibodies are directed towards antigens that are also present in the reticulocyte membrane. In the French cohort study of 265 children [4] with AIHA, reticulocytopenia with ARC below 120 × 10^9^/L was observed in 39% of the cases for a median time of 6 days (range 1–70 days). Liver function tests should also be obtained to evaluate for either the possibility of concurrent giant cell hepatitis, especially in infants and young children, or viral hepatotropic infections that may have triggered AIHA [21]. A baseline renal panel is also recommended since the disease and its treatment may affect kidney function.

Screening for underlying causes is necessary for every child presenting with w-AIHA. The evaluation should include quantification of serum immunoglobulins and antinuclear antibodies [32]. A high index of suspicion for a primary immune disorder is advised and baseline testing of lymphocyte subpopulations with flow cytometry is recommended; of note, the blood sample for this test should optimally be obtained before initiating treatment with steroids. Table 3 summarizes the initial laboratory testing of patients presenting with AIHA along with recommended evaluation for secondary causes in w-AIHA.

### 2.3. Treatment Considerations

W-AIHA tends to have a chronic course and is not expected to subside without treatment. It can be a fatal disease, with a mortality rate of up to 4% in children, either because of the acuity of the presentation or because of being refractory to treatment and requiring multiple lines of therapy with frequently associated toxicity [15].

First-line therapy starts with glucocorticoids, typically given as prednisone or prednisolone orally, although intravenous methylprednisolone may be initially needed depending on the clinical status of the patient. The dose we use is 2 to 6 mg/kg/day of prednisone or prednisone-equivalent, divided every 8–12 h; some have reported using as the initial dose up to 30 mg/kg/day of iv methylprednisolone [33]. The initial goals are decreasing hemolysis, stabilizing hemoglobin levels, and increasing safety and tolerability of packed RBC transfusion, if needed. The steroid response rate is high, up to 80%, and is usually apparent within 24 to 72 h after initiation. After normalization of hemoglobin, the steroids should be tapered slowly over approximately 6 months, since quick tapering or abrupt discontinuation have been associated with disease relapse [9].

Transfusion is frequently indicated for symptomatic or fast-progressing anemia, which can be life-threatening, especially in cases of associated reticulocytopenia, and should not be withheld just because of fear of theoretical complications [34,35]. Close communication between the treating hematologist and blood bank services will aim to determine if alloantibodies may be present, based on the history of previous transfusions or pregnancy. Prompt initiation of steroid treatment on presentation and close monitoring during transfusion is recommended to minimize the risk of transfusion reactions.

When the response to steroids in the acute setting is poor, e.g., persistent hemolysis requiring multiple transfusions 24–48 hrs after steroid initiation, we proceed to the additional first-line treatment of ivIg (1 g/kg/day × 2 days). A mixed prospective and retrospective study by Flores et al. [36] reported a positive response to ivIg in 29 of 73 patients (39.7%), including 6 out of 11 children (54%), with two variables strongly associated with a good response: the presence of hepatomegaly (with or without splenomegaly) and a low pre-treatment hemoglobin. This result is in agreement with our experience of a positive response to ivIg in children with inadequate initial response to steroids, who also frequently have a low pre-treatment hemoglobin and more severe presentation. ivIg can also be used as a therapeutic trial when DAT is negative, while the history and other laboratory data point to w-AIHA. Plasmapheresis may be rarely needed as a first-line intervention for life-threatening AIHA not responding to steroids and ivIg.

In refractory cases, when Hgb has not been stabilized over 100 g/L within 3–4 weeks post initiation of treatment, or when there is difficulty in weaning the child off steroids requiring a prednisone dose higher than 1 mg/kg/day to maintain remission [34], second-line options include rituximab (anti-CD20 antibody), splenectomy, and immunosuppressive agents. Rituximab has been used extensively as a second-line choice for refractory w-AIHA, as its effectiveness to achieve remission within 3 weeks was noted early on, in small studies [37]. Nevertheless, it is now increasingly recognized that despite initial response and remission for 1–2 years, anti-CD20 treatment for w-AIHA in children carries a significant incidence of relapse and a high risk of hypogammaglobulinemia with or without prolonged or permanent depletion of B-cells [38,39]. Intravenous immunoglobulin is typically used proactively in children as replacement therapy in hypogammaglobulinemia induced by rituximab. Rituximab is favored over splenectomy in pediatrics, with the anticipation that splenectomy will lead to life-long increased risk of thrombotic complications and sepsis by encapsulated bacteria [40]. In addition, splenectomy should be a last resort in the management of AIHA in ALPS and likely other primary immunodeficiencies, because splenectomy adds to the immunocompromised status of these patients. Indeed, overwhelming sepsis following splenectomy was shown to be the most frequent cause of death in patients with ALPS, even more frequent than malignancies [41,42].

Mycophenolate mofetil (MMF) and sirolimus are immunosuppressive agents which have a high success rate in controlling steroid-resistant AIHA cases associated with ALPS or ALPS-like syndrome. Since the challenging cases of w-AIHA in children are most commonly associated with underlying immune dysregulation syndromes (ALPS, ALPS-like disorders, primary immunodeficiencies (PID)), sirolimus is becoming the therapy of choice for long-term management of patients with relapsing w-AIHA, since it has the advantage over MMF to provide a beneficial effect towards splenomegaly [39,41,43,44]. Therefore, it is most useful to screen for underlying immunological diseases with immunoglobulin levels, lymphocyte subpopulations, and ALPS panel before commencing treatment with steroids and ivIg [32]. Children with abnormal results indicating immune dysregulation or with Evans syndrome, even if the PID has not been pinpointed genetically, are at increased risk for treatment failure and relapse and should be monitored carefully [4]. Third and fourth-line treatment strategies up to hematopoietic stem cell transplant (HSCT), which are also associated with increasing toxicity risks, have been described in previous reviews [8,34]; such escalation is rarely needed for w-AIHA in the pediatric setting.

Along with a better understanding of the PID syndromes associated with w-AIHA, new targeted therapies arise. Heterozygous cytotoxic T-lymphocyte antigen 4 (CTLA4) germline mutations, causing loss of function of this negative immune regulator, lead to a complex immune dysregulation syndrome with a high propensity for autoimmune cytopenias, along with autoimmune inflammatory lesions in the lungs, brain, and intestines [45]. Patients with mutations in the LRBA gene (encoding the lipopolysaccharide-responsive and beige-like anchor protein) have a syndrome of lymphoproliferation, humoral immune deficiency, and autoimmunity; they also show CTLA4 loss [46]. These patients frequently suffer from chronic w-AIHA or Evans syndrome, with multiple relapses. Abatacept is a soluble fusion protein comprising CTLA-4 and the Fc portion of IgG (CTLA4-Ig) inducing T regulatory cells (Treg) activity [47], that has been used successfully in the last few years to control refractory autoimmune cytopenias in patients with LRBA and CTLA-4 deficiency [46,48]. In a recent study, abatacept was administered in children with refractory AIHA following HSCT showing favorable efficacy and safety [49] and indicating that the CTLA4 pathway is likely involved in autoimmunity even without genetic mutations in the corresponding gene. More studies with novel therapeutic approaches for refractory cases of pediatric w-AIHA, using abatacept and/or other targeted treatments currently under development and in clinical trials in adults (such as the syk inhibitors), will be needed.

## 3. Cold Agglutinin Disease (CAD) due to IgM Antibody

Cold agglutinins may be seen with the primary cold agglutinin disease (CAD) or secondary cold agglutinin syndrome (CAS). The term CAD is used for chronic autoimmune hemolytic anemia due to cold-reactive IgM, typically seen in adults with a well-defined, clonal low-grade lymphoproliferative disorder of the bone marrow [50]. The term CAS refers to AIHA due to cold agglutinins, which may develop in children after infection [1].

### 3.1. Pathophysiology and Underlying Etiology

In CAS, IgM autoantibodies bind RBC membrane antigens, typically of the erythrocyte I/i antigen system, in the cold (4 °C), although they may maintain reactivity up to ≥30 °C, defined as a wide thermal amplitude. The IgM pentamers fix complement components much more readily than IgG. Complement cascade activation occurs once IgM-coated RBCs reach the core temperature and leads to the formation of the membrane attack complex C5b9 and intravascular hemolysis. C3b attached to the erythrocytic membrane also participates in extravascular clearance of the RBCs, mainly by Kupffer cells in the liver. Its cleaved product C3d is detected by DAT, producing a positive result in CAS for C3 only (Table 1). The pentavalent IgM structure is able to bind simultaneously multiple RBC resulting in agglutination and the appearance of RBC rouleaux on the blood smear.

CAS is associated with infections. Most commonly, it occurs during the course of *Mycoplasma pneumoniae* or Epstein–Barr virus (EBV) infection, but it has also been associated with other viral infections such as influenza, varicella, and measles [51,52,53]. Not all patients who develop cold agglutinins following an infection will develop clinically significant hemolysis [54]. If they do, they usually experience symptoms two weeks after the onset of the primary infection; hemolysis decreases as the infection clears up and completely resolves in eight to ten weeks.

### 3.2. Clinical and Laboratory Findings

The signs of hemolysis are similar to those already described for w-AIHA. Due to the higher rate of intravascular hemolysis, hemoglobinuria more commonly occurs in CAS rather than in w-AIHA. Additionally, there may be symptoms related to the exposure to cold temperatures, such as acrocyanosis, i.e., dark blue discoloration of distal parts of the body, such as fingers, toes, and ear lobes, is detected due to RBC agglutination and disappears upon warming. In secondary CAS, there may be signs of the primary infection, such as cough with Mycoplasma pneumoniae or hepatomegaly and increased transaminases with EBV infection.

The primary laboratory evaluation is similar for all forms of pediatric AIHA (Table 3). Hemolysis in CAS will present with low hemoglobin levels, reticulocytosis, unconjugated bilirubinemia, and increased LDH. Review of peripheral smear rarely reveals spherocytes in contrast to w-AIHA. RBC auto-agglutination and Rouleaux formation are common and suggestive of CAS.

In patients presenting with CAS, the DAT is expected to be positive for membrane-bound complement C3 and negative for IgG [55]. In DAT-positive cases, the next step would be to measure the titer of cold agglutinins in the serum; antibody titer ≥1:64 is of clinical significance [56].

Intravascular hemolysis releases high levels of hemoglobin into the bloodstream which are filtered by the kidneys. As already mentioned, this can lead to hemoglobinuria and deterioration of renal function. Thus, renal function tests, blood urea nitrogen (BUN) and creatinine, and urine dipstick examination are recommended in every child with CAS.

A serological test for Epstein–Barr virus and/or Mycoplasma pneumoniae is indicated, emphasizing IgM titers which are indicative of recent infection.

### 3.3. Treatment Considerations

Pediatric AIHA due to cold agglutinins is usually short in duration and self-limited. The management is mainly supportive, including adequate hydration and diuresis to assist with hemoglobin renal excretion. Patients with CAS are advised to avoid cold exposure until disease resolution. If there is a need for packed RBC transfusion to control life-threatening anemia, a blood warming device should be used [30]. Furthermore, addressing the underlying infection, if possible, is also one of the primary goals. Some studies support the beneficial effect of antibiotics, such as macrolides, in mycoplasma-associated hemolysis [57,58]. Steroids or other immunosuppressive medications are not indicated, and splenectomy would not be beneficial as hemolysis occurs mostly intravascularly.

Mixed AIHA with serologic characteristics of both w-AIHA and CAS are frequently associated with underlying PID and should be treated as w-AIHA.

## 4. Paroxysmal Cold Hemoglobinuria (PCH)

### 4.1. Pathophysiology and Underlying Etiology

The autoantibodies in PCH, also known as Donath–Landsteiner (DL) antibodies, are polyclonal IgG, usually directed against the erythrocytic P antigen. They bind the RBC membrane and fix complement at cold temperatures in the extremities, while complement activation upon rewarming in the body core induces intravascular hemolysis.

Historically, the first cases of PCH were described in individuals suffering from late or congenital syphilis. This form is extremely rare now and most cases present in the pediatric population, in children with a history of a common viral upper respiratory tract infection due to different common pathogens [59,60]. According to the literature, there is a male-over-female predominance (2:1) in disease frequency [61,62].

### 4.2. Clinical and Laboratory Findings

Patients typically experience a viral illness during the preceding few weeks before the appearance of hemolytic anemia. Common hemolysis manifestations include jaundice, pallor, and hemoglobinuria [62]. The onset of symptoms can be soon after cold exposure such as playing in the snow or drinking cold beverages. Thus, the recent history of such exposure can help in establishing the diagnosis. Symptoms such as fever, chills, and back and abdominal pain may also be present.

PCH evaluation should be pursued in children who have laboratory findings suggestive of intravascular hemolysis (Table 3); low hemoglobin levels, increased reticulocyte count, low haptoglobin and significantly elevated LDH, and free hemoglobin in the serum and urine can be shown. Peripheral smear may reveal polychromasia and spherocytosis [63]. The DAT is sometimes positive for C3, supporting the autoimmune etiology of hemolysis, but not for IgG, because the antibody either elutes from the cells during their preparation at room temperature or the antibody-coated cells are lysed. Negative testing should not prevent further evaluation as false-negative DAT results are common in PCH, since a special protocol has to be used for collection and preservation of the blood at 37 °C until serum isolation [5].

The Donath–Landsteiner (DL) antibody test is an in vitro biphasic assessment of the hemolytic process and is usually performed in a reference laboratory. The patient’s serum, from a blood specimen that has been maintained at 37 °C after collection, is separated from the blood cells and then incubated with donor RBCs at 4 °C. These RBCs are then transferred to 37 °C and if, when coated with DL antibody and complement, they lyse, then the test is interpreted as positive. To check for the P antigen specificity of the antibodies, ABO-compatible and P antigen-negative RBCs can be used as a negative control.

### 4.3. Treatment Considerations

PCH has a good prognosis after remission; however, it frequently presents with severe and rapidly progressive anemia which can be life-threatening. It requires supportive care, including fever control, rest, good hydration, and avoidance of cold exposure [64]. Patients with severe hemolysis may require transfusion, administered through a blood warming device. The decision is individualized based on the clinical presentation, CBC count, and rate of disease progression. Although PCH typically does not require treatment with steroids, it is not a mistake to start a regimen similar to that used for w-AIHA and just discontinue after a few days what diagnosis is established and the patient has improved. Any primary condition contributing to the disease should be addressed as well.

## Figures and Tables

**Figure 1 jcm-10-00216-f001:**
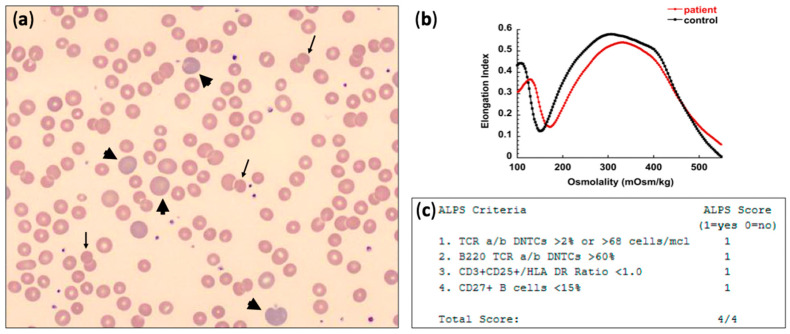
(**a**) Peripheral blood smear showing polychromasia (arrowheads) and occasional spherocytes and microspherocytes (arrows); (**b**) osmotic gradient ektacytometry in w-AIHA may be normal or compatible with acquired spherocytosis as in this case: showing mildly increased Omin and decreased EImax; (**c**) autoimmune lymphoproliferative syndromes (ALPS) screening panel positive for 4 out of 4 criteria.

**Table 1 jcm-10-00216-t001:** Classification of autoimmune hemolytic anemia (AIHA).

Type	Antibody Class	T of Maximal Reactivity	DAT Positivity
**Warm antibody AIHA (w-AIHA)**	IgG	37 °C	IgG ± C3
**Cold agglutinin syndrome (CAS)**	IgM	4 °C	C3 only
**Mixed AIHA**	cold IgM and warm IgG	4 °C and 37 °C	IgG and C3
**Paroxysmal cold hemoglobinuria (PCH)**	IgG	4 °C	±C3

T: Temperature (°C); IgG: immunoglobulin G; IgM: immunoglobulin M; DAT: direct agglutinin test; C3: complement component 3.

**Table 2 jcm-10-00216-t002:** Differential diagnosis of hemolysis in children.

**HEREDITARY HEMOLYTIC ANEMIAS**
**Membrane defects** Hereditary spherocytosis Hereditary elliptocytosis and pyropoikilocytosis Southeast Asian ovalocytosis Dehydrated hereditary stomatocytosis or hereditary xerocytosis RBC Overhydration syndromes
**Enzymopathies** Glucose-6-phosphate dehydrogenase (*G6PD*) deficiency Pyruvate kinase (*PKLR*) deficiency Other RBC enzyme disorders (*AK1*, *ALDOA*, *GCLC*, *GPI*, *GPX1*, *GSR*, *GSS*, *HK1*, *NT5C3A*, *PFKM*, *PGK1*, *TPI1*)
**Hemoglobin disorders** Sickle cell disease Thalassemias Unstable hemoglobins
**Congenital dyserythropoietic anemias**
**ACQUIRED HEMOLYTIC ANEMIAS**
**Autoimmune hemolytic anemia (AIHA)** Warm-reactive AIHA Cold agglutinin syndrome Paroxysmal cold hemoglobinuria (PCH) Drug-induced (very rare in children)
**Alloimmune hemolytic anemia** Neonatal alloimmune hemolysis Post-transfusion hemolysis Acute hemolytic reactions Delayed hemolytic reactions
**Traumatic Hemolytic Anemia** Impact Macrovascular defects-prostheses (Waring blender syndrome with a dysfunctional mechanical heart valve) Microvascular Typical and Atypical Hemolytic uremic syndrome Thrombotic thrombocytopenic purpura Disseminated intravascular coagulation
**Hypersplenism**
**Hemolytic Anemia due to toxic effects on the membrane** Spur cell anemia in severe liver disease External toxins Animal or spider bites Metals Organic compounds Infection
**Paroxysmal nocturnal hemoglobinuria**

**Table 3 jcm-10-00216-t003:** Diagnostic evaluation of children with AIHA.

**Initial Laboratory Evaluation of AIHA**
Complete blood count (CBC) with differential Reticulocyte count (absolute reticulocyte count (ARC) is preferable) Peripheral blood smear review
Direct antiglobulin test (DAT) Type and Screen [Screen is performed by the Indirect Antiglobulin Test (IAT)] *Follow-up evaluations include:* Cold agglutinin titer when DAT is positive for C3 Donath-Landsteiner antibody test when paroxysmal cold hemoglobinuria (PCH) is suspected Enhanced DAT (Super-Coombs test) if presentation is consistent with AIHA but conventional DAT is negative
Serum markers of hemolysis (i.e., total and unconjugated bilirubin, lactate dehydrogenase, haptoglobin) Liver and kidney function tests
Urine hemoglobin and hemosiderin evaluation may be used to differentiate intravascular (positive result) versus extravascular hemolysis
Bone marrow aspirate and biopsy in atypical cases where there is a concern for underlying malignancy, e.g., concurrent thrombocytopenia and/or neutropenia, unusual or prolonged reticulocytopenia, lymphadenopathy or organomegaly without evidence of concurrent EBV infection.
**In cases of w-AIHA, consider possibility for underlying causes**
*Screen for primary immune disorder (PID) on a sample obtained before treatment initiation* IgG, IgM, IgA quantification Lymphocyte subpopulations by flow cytometry Autoimmune lymphoproliferative syndrome (ALPS) screening panel by flow cytometry *Follow-up testing as needed with next-generation sequencing on ALPS or PID gene panels*
*Screen for rheumatologic diseases (frequently indicated in teenager females)* Antinuclear antibodies Anti-double-stranded DNA antibodies
HIV testing

## Data Availability

Data sharing not applicable.

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
