# Peer review of "Autoimmune Hemolytic Anemia in the Pediatric Setting"

_jcm, 2021, doi:10.3390/jcm10020216_

Round 1
Reviewer 1 Report
The authors present a very readable account of Autoimmune Hemolytic Anemia in the Pediatric setting. They cover both major and rare types of AIHA in sufficient detail to leave the reader informed and what investigations should be undertaken.
Major Comments:
The manuscript may benefit from a table summarizing which laboratory investigations should be performed in all cases, together with secondary investigations that are helpful in the differentiation the rarer types and any underlying causes. It seems to me that given the rarity of pediatric AIHA then identifying or excluding immunodeficiency should be undertaken in all cases.
Minor Comments:
Line 84: the hemoglobin value should be 73 g/L
Figure 1a: Some of the arrowheads do not correspond with a cell type.
Author Response
Response to Reviewer 1
We are grateful for your positive review and constructive comments. We have implemented the recommended changes in our revised manuscript which we feel have improved the quality of our paper.
Major Comment: The manuscript may benefit from a table summarizing which laboratory investigations should be performed in all cases, together with secondary investigations that are helpful in the differentiation of the rarer types and any underlying causes. It seems to me that given the rarity of pediatric AIHA then identifying or excluding immunodeficiency should be undertaken in all cases.
We thank the reviewer for this important comment and for the opportunity to elaborate further. We agree that at least screening for underlying immunodeficiency should be done for all pediatric cases of w-AIHA on presentation, followed by further evaluation as directed by these results or if the disease is refractory or persistent/recurrent. A table summarizing the diagnostic workup for new AIHA cases in children is now included in our manuscript (Table 3).
Minor Comments:
Line 84: the hemoglobin value should be 73 g/L
Figure 1a: Some of the arrowheads do not correspond with a cell type.
We thank the reviewer for those corrections. We have edited the manuscript accordingly.
Reviewer 2 Report
Very good review by experts in the field.
I have only a few comments.
Major one: the differential diagnosis is not explicated. This may be due to the lack of room given. Nevertheless this is not so easy for non-hematologists. Maybe a Table with a list of main diagnosis to be considered in the pediatric setting, either constitutional (especially G6PD deficiency) and acquired (HUS & thrombotic microangiapathy, PNH,…), and some clues for diagnosis will be of help.
Minor comments:
p2: the frequency given for DAT-negative cases is « up to 11% » : does this refer to a study in children ?
p5: reticulocytopenia may be seen before reticulocytes response but, as underlined by the authors, may also be due to antibodies directed against antigens present on reticulocytes. In some cases actually reticulocytopenia is associated with erythroblastopenia and in such patients the reticulocytes response is much more delayed with eventually prolonged need for transfusion support. This may be demonstrated by bone marrow aspiration analysis. What are your indications for BMA in this setting?
p6: we agree to give steroids IV at least in every child with severe presentation and until the reticulocytes response occurs (not only for 2 days).
p6: the dose of steroids should be given
p6: do you have a reference for the benefit associated with ivIg in this pediatric setting? If not precise "in our hands"
p6: a few patients are steroid primary refractory. What is your definition for steroid resistance and introduction of a second-line treatment? Our pragmatic definition is “transfusion need after 2 weeks of adequate steroid treatment”.
p6: splenectomy in AIHA may also be not adequate if you consider the risk for underlying immune deficiency. Please comment.
Author Response
Response to Reviewer 2
We are grateful for your positive review and constructive comments. We have implemented the recommended changes in our revised manuscript which we feel have improved the quality of our paper.
Major one: the differential diagnosis is not explicated. This may be due to the lack of room given. Nevertheless, this is not so easy for non-hematologists. Maybe a Table with a list of main diagnosis to be considered in the pediatric setting, either constitutional (especially G6PD deficiency) and acquired (HUS & thrombotic microangiopathy, PNH,…), and some clues for diagnosis will be of help.
We appreciate the reviewer’s recommendation. We have now included a table (Table 2) to provide the range of differential diagnosis for hereditary and acquired hemolytic anemias that needs to be considered when a child presents with hemolysis/hemolytic anemia. The direct antiglobulin test (DAT) differentiates autoimmune hemolytic anemia from other causes of hemolysis, although exceptions exist as detailed in our manuscript.
Minor comments:
p2: the frequency given for DAT-negative cases is « up to 11% » : does this refer to a study in children ?
Indeed, the references 5 and 9 cited in our manuscript are studies reporting such frequency of DAT-negative results in pediatric populations. Reference 5 reports negative DAT in 5 out of 64 patients with w-AIHA (7.8%) and reference 9 reports negative DAT in 3 out of 26 patients with w-AIHA (11%)
- Vaglio, S.; Arista, M.C.; Perrone, M.P.; Tomei, G.; Testi, A.M.; Coluzzi, S.; Girelli, G. Autoimmune hemolytic anemia in childhood: serologic features in 100 cases. Transfusion 2007, 47, 50-54, doi:10.1111/j.1537-2995.2007.01062.x.
- Naithani, R.; Agrawal, N.; Mahapatra, M.; Kumar, R.; Pati, H.P.; Choudhry, V.P. Autoimmune hemolytic anemia in children. Pediatr Hematol Oncol 2007, 24, 309-315, doi:10.1080/08880010701360783.
p5: reticulocytopenia may be seen before reticulocytes response but, as underlined by the authors, may also be due to antibodies directed against antigens present on reticulocytes. In some cases actually reticulocytopenia is associated with erythroblastopenia and in such patients the reticulocytes response is much more delayed with eventually prolonged need for transfusion support. This may be demonstrated by bone marrow aspiration analysis. What are your indications for BMA in this setting?
That is a very good point. We do perform and recommend bone marrow studies in atypical cases of autoimmune hemolytic anemia where there is any concern for underlying malignancy. Such concern may be due to concurrent thrombocytopenia and/or neutropenia, unusual or prolonged reticulocytopenia, lymphadenopathy or organomegaly without evidence of concurrent EBV infection. We have now added this statement in Table 3 listing recommended laboratory evaluations for AIHA.
p6: we agree to give steroids IV at least in every child with severe presentation and until the reticulocytes response occurs (not only for 2 days).
We have changed this statement to: “intravenous methylprednisolone may be initially needed depending on the clinical status of the patient”, deleting “for the first couple of days”. We do change iv methylprednisolone to oral prednisone or prednisolone when the patient is clinically well and able to take oral medications since the bioavailability of oral steroids is excellent.
p6: the dose of steroids should be given
We have now added the steroid dose we use: “2 to 6 mg/kg/day of prednisone or prednisone-equivalent, divided every 8-12 hours” and have added that “some have reported to use as initial dose up to 30 mg/kg/day of iv methylprednisolone [33]”.
We have found that adding ivIg has been more effective for our refractory cases than increasing the dose of methylprednisolone from 6 mg/kg/day to 30 mg/kg/day.
p6: do you have a reference for the benefit associated with ivIg in this pediatric setting? If not precise "in our hands"
We cite now Flores et al [36] who reported a positive response to ivIg in 29 of 73 patients (39.7%), including 6 out of 11 children (54%), with two variables strongly associated with a good response: the presence of hepatomegaly (with or without splenomegaly) and a low pre‐treatment hemoglobin. This result is in agreement with our experience of a positive response to ivIg in children with inadequate initial response to steroids, who also frequently have a low-pretreatment hemoglobin and more severe presentation.
p6: a few patients are steroid primary refractory. What is your definition for steroid resistance and introduction of a second-line treatment? Our pragmatic definition is “transfusion need after 2 weeks of adequate steroid treatment”.
We plan to start a second-line treatment if we do not see stabilization of Hb to >10 g/dL (100 g/L) within 3-4 weeks after treatment initiation or if the patient does not tolerate tapering of glucocorticoids and require a prednisone dose higher than 1 mg/kg/day to maintain remission [34]. We have now added this clarification in the lines 265-267.
p6: splenectomy in AIHA may also be not adequate if you consider the risk for underlying immune deficiency. Please comment.
We thank the reviewer for the comment. We agree it is important to note that splenectomy should be a last resort in the management of AIHA in ALPS because splenectomy adds to the immunocompromised status of these patients. Indeed, overwhelming sepsis following splenectomy was shown to be the most frequent cause of death in patients with ALPS, even more frequent than malignancies [41,42].
41. Rao, V.K. Approaches to Managing Autoimmune Cytopenias in Novel Immunological Disorders with Genetic Underpinnings Like Autoimmune Lymphoproliferative Syndrome. Front Pediatr 2015, 3, 65, doi:10.3389/fped.2015.00065.
42. Price, S.; Shaw, P.A.; Seitz, A.; Joshi, G.; Davis, J.; Niemela, J.E.; Perkins, K.; Hornung, R.L.; Folio, L.; Rosenberg, P.S., et al. Natural history of autoimmune lymphoproliferative syndrome associated with FAS gene mutations. Blood 2014, 123, 1989-1999, doi:10.1182/blood-2013-10-535393.